# Driving Effect of Decoupling Provincial Industrial Economic Growth and Industrial Carbon Emissions in China

**DOI:** 10.3390/ijerph20010145

**Published:** 2022-12-22

**Authors:** Jingfen Hua, Junli Gao, Ke Chen, Jiaqi Li

**Affiliations:** 1Key Lab of Urban Environment and Health, Institute of Urban Environment, Chinese Academy of Sciences, Xiamen 361021, China; 2Research Center of Urban Carbon Neutrality, Institute of Urban Environment, Chinese Academy of Sciences, Xiamen 361021, China; 3Shenzhen Academy of Environmental Sciences, Shenzhen 518000, China; 4Finance and Statistics College, Hunan University, Changsha 410006, China; 5Longcheng Street Office, Longgang District, Shenzhen 518100, China

**Keywords:** industrial growth, carbon emissions, velocity decoupling, quantity decoupling, panel regression analysis

## Abstract

China is facing the dual challenges of fostering economic growth and mounting an effective response to climate change, so it is vital to continue promoting industrial carbon emission reduction. This paper uses panel data from 1998 to 2019 to measure the industrial carbon emissions of 30 provinces in China. The Tapio decoupling and IPAT (Impact = Population × Affluence × Technology)-based decoupling models are used to analyze each province’s velocity and quantity decoupling index for industrial carbon emissions. The fixed effect model analyzes the influencing factors for carbon decoupling. The results show that the industrial carbon emissions of various provinces in China are increasing yearly, but there are significant differences among provinces. The carbon decoupling of the industrial economy in most provinces is weak, and the quantitative decoupling index is better than the velocity decoupling index. The cleanliness of energy, balance, and labor productivity significantly affect the velocity decoupling index. The cleanliness of energy, the industry’s structure, and the population significantly affect the quantity decoupling index. Based on empirical results, the study puts forward some policies to promote the efficient carbon decoupling of the industrial economy.

## 1. Introduction

Since its reform and expansion, China’s industrial economy has proliferated, and China became the world’s top manufacturing country in 2010. China also became the only country in the world among all the categories in the United Nations Industrial Classification with a complete industrial system. China’s industrial value added increased 43.73 times in 2019 compared to 1978 at comparable prices. However, China’s high industrial growth has come at the cost of high energy consumption and high CO_2_ emissions. The growth of the industrial economy has been accompanied by a significant increase in China’s total carbon emissions, and China became the world’s top carbon emitter in 2005. In 2019, China accounted for 29.4% of total global CO_2_ emissions [1], with industrial CO_2_ emissions accounting for 79.3% of all emission sources [2]. As the first developing country to submit an autonomous national contribution, China has committed to achieving carbon peaking in 2030 and strives to achieve carbon neutrality by 2060 [3]. Compared with some developed countries whose emissions have peaked, China’s carbon emissions are still in the growth period. Achieving carbon neutrality in such a short period is a massive challenge for China. Therefore, it is necessary and urgent to break the inevitable link between “high industrial economic growth and high CO_2_ emissions” to study the sustainable development strategy of Chinese industry.

At present, the research on the relationship between industrial economic growth and carbon emissions is mainly focused on the following aspects: (1) using the environmental Kuznets curve to study the relationship between industrial growth and carbon emissions [4]; (2) testing the causal relationship between industrial economic growth and carbon emissions by econometric means [5,6]; (3) quantitative analysis of the related factors of carbon emissions by exponential decomposition method [7,8]; and (4) constructing a decoupling index to measure the relationship between industrial economic growth and environmental problems [9,10,11]. Among them, decoupling has attracted scholarly attention as an essential indicator of the successful integration of the economy and the environment. Its essence is to measure whether economic growth is at the cost of resource consumption and environmental damage. Carbon decoupling refers to the relationship between CO_2_ emissions and changes in economic growth. A negative growth or slowdown in CO_2_ emissions can be seen as decoupling when the economy grows. 

From the perspective of decoupling measurement methods, the existing literature mainly uses four methods to study the decoupling relationship between economic growth and carbon emissions: (1) the OECD (Organization for Economic Co-operation and Development) decoupling index model includes absolute decoupling and relative decoupling [12]; (2) the Tapio decoupling model introduces the idea of elastic decoupling into the OECD decoupling model [13]; (3) the environmental Kuznets curve model [14]; and (4) a decoupling model based on the IPAT equation [15]. The above four decoupling analysis methods can be divided into two categories: velocity decoupling analysis and quantitative decoupling analysis. The carbon emission velocity decoupling model quantifies the decoupling state through the relationship between the CO_2_ emission change rate and economic growth rate, including the OECD decoupling model and the Tapio decoupling model. Compared with the OECD decoupling model, the Tapio decoupling model has become the most widely used decoupling model because of its advantages of being insensitive to the choice of the base period and not being affected by differences in statistical dimensions [16]. Unlike velocity decoupling, quantitative decoupling focuses on quantitative relationships, meaning that carbon emissions tend to stabilize or decrease with economic growth, mainly including the EKC model and the decoupling model based on the IPAT equation. Thus, scholars generally believe that the carbon intensity target is a total carbon emission control target [17]. 

In the studies, scholars mainly applied one of the above decoupling models to explore the decoupling state between economic growth and carbon emissions. Freitas and Kaneko [18], and Xiao et al. [19] used the decoupling model to analyze the long-term decoupling state of economic growth and carbon emissions in Brazil, China, and the United States. Wu et al. [20], Wang et al. [21], Li et al. [22], and Liu et al. [23] measured the decoupling state of economic growth and carbon in Beijing, Shaanxi, Shanxi, and Jiangxi provinces of China, respectively. A large number of studies have shown that there is a close relationship between industrial activities and carbon decoupling, and the final energy consumption of the industrial sector has become carbon-intensive. Liu et al. [24] analyzed China’s industrial sector, using the decomposition method from 1998 to 2005, and found that industrial activity and energy intensity factors were dominant contributors to the change in CO_2_. Zhao et al. [25] employed decomposition analysis for industrial emissions in Shanghai city from 1996 to 2007 and found that energy intensity and industrial structure were key factors in reducing CO_2_. Wen et al. [26] found similar results for 38 sub-industries in China from 2000 to 2017 and estimated that investment, output, and energy scale were the major contributors to raising CO_2_. Wang et al. [27] analyzed the data of various industries in the United States from 1997 to 2016. Their investigation showed that the industrial sector was the main factor in the rise in CO_2_. Therefore, the further reduction of carbon intensity must depend on the efforts of the industrial sector [28,29]. 

Some scholars have also studied carbon decoupling for the industrial economy. Gong et al. [30], Huang et al. [31], and Yue et al. [32] analyzed the carbon decoupling status of the industrial economy in Wuhan, the Yangtze River Economic Zone, and the western region, respectively. Yu [33], Jie et al. [34], and Lv [35] analyzed the carbon decoupling status of China’s power industry, energy extraction industry, and commerce and distribution industry, respectively, using decoupling models.

However, the decoupling analysis method can only reflect the dependency relationship between the two and cannot explore the reasons for the said dependency relationship. To break through this limitation, some scholars have studied the driving mechanism of decoupling by combining the decoupling model with the decomposition method. Many scholars researched the decoupling model, Log-averaged Dependent Decomposition Index (LMDI) model, and the panel regression model. Yuan et al. [36] and Zhang et al. [37] studied the internal mechanism of decoupling from the perspective of the whole country. Lv et al. [38] and Wang [39] explored the factors influencing the decoupling relationship between carbon emissions and economic growth in one or several provinces.

In summary, the current research has mostly used panel data to analyze economic carbon decoupling, and the results have provided some theoretical support and reference for further research. However, there is still room for improvement in related research. Among the research perspectives on the decoupling of economic growth and carbon emissions, many scholars have examined the decoupling status and internal driving mechanisms for the whole of China or a few provinces and cities. Since carbon emission reduction targets require the joint efforts of all provinces, it is necessary to investigate the decoupling status of all provinces to develop tailor-made emission reduction measures for different provinces. From the perspective of decoupling industrial economic growth and carbon emissions, some scholars have examined a limited number of industrial sectors. However, not all industrial sectors have been included in the study. From the perspective of the carbon decoupling driving mechanism, most scholars have only considered a limited number of driving indicators. They have not considered external drivers such as the economy, energy, innovation, and productivity. In addition, no scholars have been found who have combined velocity decoupling and quantity decoupling to analyze industrial economic growth and carbon emissions.

Based on the above, this paper adopts the Tapio decoupling model and the decoupling model based on the IPAT equation to measure the industrial production in China (Tibet, Hong Kong, Macau, and Taiwan are not discussed due to the lack of data) for 22 consecutive years from 1998 to 2019. The two models are based on incremental data and the IPAT equation. Moreover, this paper adopts the fixed effect model to analyze the effectiveness of the driving factors of velocity decoupling and quantitative decoupling in 30 provinces in China from 2003 to 2019—for 17 consecutive years after China approved the Kyoto Protocol of the United Nations Framework Convention on Climate. Finally, according to the results, this paper put forward relevant countermeasures from the perspective of government and industrial companies.

## 2. Materials and Methods

### 2.1. Study Area

In this study, 30 provinces in China were selected as study areas. The paper will use the word “province” to refer to these study areas throughout the rest of the paper. The research period of industrial decoupling state accounting was from 1998 to 2019 (the carbon dioxide data for some provinces were missing before 1998.) The research period of industrial carbon decoupling driver analysis was from 2003 to 2019 (The year 2003 was the first year after China approved the Kyoto Protocol to the United Nations Framework Convention on Climate change, and it was a key time point for China to carry out carbon dioxide emission reduction).

### 2.2. Research Hypothesis

Chinese industrial enterprises face two major demands: high economic growth and carbon emission reduction. However, the demand for economic development inevitably leads to an increase in energy consumption, increasing CO_2_ emissions, which runs counter to the demand for carbon emission reduction in industrial enterprises. Under the premise of economic development, promoting carbon emission reduction in industrial enterprises has become an important breakthrough point to resolving the above contradictions. Industrial enterprises and the government will implement measures to balance the “economy and being green.” The transmission mechanism for decoupling the industrial economy from carbon is shown in Figure 1.

As shown in Figure 1, at the national level, the government can adjust the structure of industrial industries through economic measures, such as tax reduction, loan subsidies, export credits, and other policy support, in order to reduce the proportion of resource-intensive industries and increase the proportion of capital-intensive and labor-intensive industries, thereby reducing carbon dioxide emissions. At the same time, the government can also strengthen the investment in scientific research and increase the intensity of investment in science and technology in industries to achieve the goal of reducing carbon dioxide emissions to ensure economic growth. However, if a company does not meet the carbon market threshold to trade in the carbon market, it will reach its emission reduction needs in other ways. The first way is to adjust the energy structure to increase the proportion of clean energy in the energy consumption structure. “Clean” energy here is relative, depending on the carbon content of the energy source. For example, the carbon content of coal is about 30% higher than that of oil and about 70% higher than that of natural gas [40]. The second way is to improve the balance of the energy consumption structure, diversify energy consumption varieties, and break the dilemma that energy consumption depends on coal. Industrial enterprises can also achieve their emission reduction needs by improving their labor–output ratio through training and upgrading human and material resources to reduce CO_2_ emissions for the same output unit.

In summary, from a theoretical point of view, reforming the industrial structure, improving scientific and technological inputs, reforming the energy structure, and improving the labor–output ratio can all drive the level of carbon decoupling in industrial economies to some extent.

It is hard to simply measure whether the intrinsic drivers of the industrial carbon decoupling effect are effective using a theoretical analysis. It is necessary to quantitatively and empirically analyze whether the intrinsic drivers of the industry, such as the energy consumption structure, industrial structure, technology input intensity, and labor productivity, can effectively enhance carbon decoupling. 

**Hypothesis** **H1:**
*When the industrial energy consumption structure tends to be cleaner, the level of industrial carbon decoupling increases.*


**Hypothesis** **H2:**
*When the industrial energy consumption structure tends to diversify, the level of industrial carbon decoupling increases.*


**Hypothesis** **H3:**
*When industrial industry structure tends to be non-resource-intensive, the level of industrial carbon decoupling increases.*


**Hypothesis** **H4:**
*When the labor–output ratio of industrial enterprises increases, the level of industrial carbon decoupling increases.*


**Hypothesis** **H5:**
*When the ratio of R&D input from local governments increases, the level of industrial carbon decoupling increases.*


**Hypothesis** **H6:**
*When investment in the environment increases, the level of industrial carbon decoupling increases.*


**Hypothesis** **H7:**
*When the regulation strength of local governments increases, the level of industrial carbon decoupling increases.*


**Hypothesis** **H8:**
*When the urbanization ratio increases, the level of industrial carbon decoupling increases.*


### 2.3. Data Sources and Descriptions

(1) Industrial added value: Industrial added value is obtained from provincial and municipal statistical yearbooks. In order to eliminate the influence of price factors on the study, this paper takes 1995 as the base period and applies the deflator to the industrial ex-factory price index to obtain the actual industrial added value in all years.

(2) Carbon emissions of the industry: The data come from Carbon Emission Accounts & Data. Using the IPCC sector emission accounting method [41], 41 industrial sectors and 17 kinds of fossil fuels in China’s energy system are selected, and the fuels consumed in thermal energy and thermal energy production are considered [42,43]. According to the formula “carbon emissions = energy consumption × emission factors,” the carbon dioxide produced by fossil fuel combustion in different industrial sectors is obtained. Finally, the carbon emissions of different industrial sectors are summed up, and the industrial carbon emissions of provinces and cities are calculated.

(3) Cleanliness of energy: $EC=\frac{g_{3}}{g_{t}}$. According to the relative principle, this paper selects natural gas and oil consumption, which contain less carbon, to represent clean energy, and uses the ratio of their consumption to industrial energy consumption as a measure of the cleanness of industrial energy consumption. The larger the indicator is, the higher the degree of cleanliness of the industrial energy consumption structure.

(4) Balance of energy: $EB=\sum_{k=1}^{m}\frac{\ln(g_{k}/g_{t})}{\ln m}\cdot \frac{g_{k}}{g_{t}}$. In this paper, we construct the formula for the industrial energy consumption balance degree based on information entropy, which is the ratio of information entropy to maximum entropy. m refers to the number of industrial energy consumption categories. Because in this paper, coal, crude oil, and natural gas will be recorded as the primary energy species, *m =* 3; $g_{k}/g_{t}$ represents the percentage of energy *k* consumption out of the total energy consumption $g_{t}$ after the calorific value is converted into standard coal. The energy balance represents the difference between the proportions of industrial energy consumption. The indicator is between 0 and 1; the larger the indicator, the more balanced the energy consumption.

(5) Productivity of labor: $LP=\frac{\mathrm{industrial}\;\mathrm{value}-\mathrm{added}}{\mathrm{average}\;\mathrm{number}\;\mathrm{of}\;\mathrm{workers}\;\mathrm{in}\;\mathrm{industrial}\;\mathrm{sector}}$. The average number of laborers includes the average number in both state-owned and private enterprises. The larger the indicator, the higher the labor productivity in the industry.

(6) Structure of industry: $SIO=\frac{\mathrm{Value}\;\mathrm{added}\;\mathrm{of}\;\mathrm{resource}-\mathrm{intensive}\;\mathrm{industries}}{\mathrm{Value}\;\mathrm{added}\;\mathrm{of}\;\mathrm{whole}\;\mathrm{industry}}$. This paper draws on the classification standard of the United Nations SITC and related research [44] and defines it as the ratio of the value added of resource-intensive industries to the value added of the whole industrial industry. The smaller indicator represents a relative decrease in the share of value added of resource-intensive industries. It also represents the existence of economic actions of the government or enterprises that lead to a tilt in the structure of industrial industries toward non-resource-intensive industries, which may lead to a decrease in industrial carbon emissions.

(7) The intensity of technology investment: $STS=\frac{R\&D\;\mathrm{spent}}{\mathrm{GDP}}$. This refers to the proportion of GDP actually spent on basic research, applied research, and experimental development by the society during the year. The higher the society-wide spending on science and technology-related expenditures, the more likely it is to have a spillover effect on improving science and technology levels in industries.

(8) Increase environmental investment: This refers to the local government’s annual industrial investment. The higher the index level, the lower local industrial carbon emissions.

(9) Number of population: This term refers to population figures from year-end statistics for each province. Theoretically, carbon emissions also increase when the population number increases and social activity increases.

(10) Urbanization ratio: $\frac{\mathrm{Urban}\;\mathrm{Population}}{\mathrm{Number}\;\mathrm{of}\;\mathrm{Population}}$. When a province’s urbanization ratio increases, the degree of industrial agglomeration increases, and the industrial carbon emissions decrease.

(11) Government regulation: $\sum {\mathrm{SO}}_{2}+\mathrm{PM}$. This refers to the degree of government regulation of industrial carbon emissions. Carbon dioxide and air pollutants are “homogenous,” and industrial production emits carbon dioxide and air pollutants. Therefore, many scholars consider the emissions of air pollutants as the level of government environmental regulation of the industry.

The descriptive statistics of the above variables are shown in Table 1.

### 2.4. Calculation Method of Industrial Carbon Emission Velocity Decoupling Index

Tapio index is not sensitive to the selection of the base period, nor is it affected by differences in statistical dimensions. In order to fully reflect the volatility of the dynamic data decoupling state and the objective accuracy of decoupling prediction, this paper chooses the Tapio decoupling model combined with incremental data to calculate the industrial carbon emission velocity decoupling index. The formula is shown in (1):(1)$$D_{s}=\frac{\frac{\Delta I}{I_{0}}}{\frac{\Delta G}{G_{0}}}=\frac{\frac{I_{n}-I_{0}}{I_{0}}}{\frac{G_{n}-G_{0}}{G_{0}}}$$
where *D_s_* is the velocity decoupling index between industrial carbon emissions and industrial economic growth, *I_n_* and *G_n_* are the current values of industrial carbon emissions and industrial value added, respectively, and *I*_0_ and *G*_0_ are the base period values of industrial carbon emissions and industrial added value, respectively.

The decoupling states are divided into 8 categories according to the size of ∆*I/I*_0_, ∆*G/G*_0_, and the velocity decoupling elasticity, as shown in Table 1.

### 2.5. Calculation Method of Industrial Carbon Emission Quantitative Decoupling Index

A decoupling state can be considered to occur only when there is an increase in the total amount of the economy and a flat or decreasing amount of environmental pollution. Lu et al. [45,46,47,48] proposed a functional relationship between the rate of change in carbon emission intensity and the GDP growth rate based on the IPAT equation [49] to investigate the quantitative decoupling of economic growth and carbon emission. Some scholars believe that quantitative decoupling can be a more practical measurement method as it can provide more guidance in environmental resource protection.

Based on the IPAT equation and referring to the relevant studies of Lu, the formula for calculating the quantitative decoupling index of industrial carbon emissions is derived:(2)$$I_{n}=P_{n}\times A_{n}\times T_{n}=G_{n}\times T_{n}=G_{0}\times {\left(1+g\right)}^{n}{\times T}_{0}{\times (1-t)}^{n}=G_{0}\times T_{0}\times {\left[\left(1+g)\times (1-t\right)\right]}^{n}$$
where *I_n_*, *P_n_*, *A_n_*, *T_n_*, and *G_n_* are current industrial carbon emissions, population, per capita industrial added value, industrial carbon emission intensity and industrial added value, respectively. *I*_0_, *T*_0_, and *G*_0_ are the base period values of industrial carbon emissions, industrial carbon emissions intensity, and industrial added value, respectively.

In order to realize the absolute decoupling of industrial carbon emissions from industrial economic growth, it is necessary to meet the annual increase of industrial carbon emission intensity *k_r_* ≤ 0. According to the nature of the inequality, we derive (1 + *g*)*^n^* × (1 − *t*)*^n^* ≤ 1. It is known that the relationship between *t* and *g* is very important, so the ratio of *t* to *g* is taken as the quantitative decoupling index, which is recorded as *D_q_*. The decoupling states are divided into 8 categories according to the size of *g*, *t*, and the quantity decoupling elasticity, as shown in Table 2.

### 2.6. Method for Calculating the Effectiveness of Industrial Carbon Decoupling Drivers

Panel data regression models are well-suited for factor analysis. First, panel data can analyze comprehensive factors. Because panel data regression includes both cross-sectional and temporal dimensions, it can solve the problems that separate cross-sectional and temporal data cannot solve. For example, cross-sectional data regression cannot observe the effect of labor productivity over time. Second, the panel data can expand the number of samples and increase the stability and accuracy of the model regression results. Third, when conducting factor analysis, panel data fixed effects regression can solve the problem of factor omission. When factors are omitted in the panel data regression, the unobserved factors are differenced out when demeaning. This reduces the endogeneity problem due to unobserved factors that may correlate with the explanatory variables [50,51].

Therefore, this paper used two panel data regression models to study the driving effect of decoupling between provincial industrial economic growth and industrial carbon emissions in China.

The decoupling index rate is the industrial CO_2_ emissions required to increase 1 unit of industrial value added. The quantity decoupling index is the rate of decrease in industrial carbon intensity that accompanies a 1 percentage point increase in industrial value added. A combination of factors determines the decoupling index. Therefore, this paper establishes a regression model, as shown in Equation (3).
(3)$$D_{it}=\alpha_{it}+\beta_{1}EB_{it}+\beta_{2}EC_{it}+\beta_{3}LP_{it}+\beta_{4}SIO_{it}+\beta_{5}STS_{it}+\beta_{6}IIE_{it}+\beta_{7}NP_{it}+\beta_{8}UR_{it}+\beta_{9}GR_{it}\mu_{it}$$

In this model, $D_{it}$ represents the velocity decoupling index or quantity decoupling index between industrial economic growth and industrial carbon emissions in a province, period t; $EB_{it}$ is the energy balance in province i, period t; $EC_{it}$ is the energy cleanliness in province i, period t; $LP_{it}$ is the labor productivity in province i, period t; $SIO_{it}$ is the industrial industry structure in province i, period t; $STS_{it}$ is the science and technology input intensity in province i, period t; $IIE_{it}$ is the increase in environmental investment in province i, period t; NP is the number of population in province i, period t; $UR_{it}$ is the urbanization ratio in province i, period t; $GR_{it}$ is the government regulation in province i, period t; and $\mu_{it}$ is a random perturbed variable.

## 3. Results 

### 3.1. Analysis of Decoupling and Carbon Emissions in China’s Provinces

#### Accounting Results and Analysis of Industrial Carbon Decoupling in Different Provinces of China

The average industrial value added of China’s provinces grew from 100,638.1 million yuan in 1997 to 866,588.4 million yuan in 2019, with a general upward trend from 1997 to 2017, and a slight downward trend after 2016. The average industrial carbon emissions increased from 0.817 billion tons in 1997 to 310.21 million tons in 2019, with the growth rate of industrial carbon emissions showing a decreasing trend year by year since 2003, and a slightly increasing trend since 2015. The average industrial value added and industrial carbon emissions of each province in China show regional differences.

Generally, China can be divided into eight economic zones. The industrial structure and development level of each economic zone are different, resulting in obvious regional differences in industrial added value and industrial carbon emissions. (See Figure 2 and Figure 3 for details.) The most influential and multifunctional manufacturing center on the Eastern coast and the manufacturing center of high-tech products on the southern coast always have the highest average industrial added value, and the average industrial carbon emissions on the eastern coast are also high. Northwest is an important energy strategic replacement base, and the tertiary industry in the southwest is developing rapidly. These two economic zones are both vast and sparsely populated, and their economic development conditions are poor, so the average industrial added value is the lowest, and the average industrial carbon emissions are always low. The middle reaches of the Yellow River are the largest coal mining and deep processing base, natural gas and hydropower development base, iron and steel industry base, and non-ferrous metal industry base. The northern coastal area is one of the most powerful high-tech R&D and manufacturing centers. The average industrial carbon emissions of these two economic zones are always high. The northeast is the manufacturing base of energy and raw materials, with large energy consumption but a low level of economic development, so the average industrial added value is smaller, but the average industrial carbon emissions are higher. The middle reaches of the Yangtze River are a raw material base dominated by iron and steel and non-ferrous metallurgy, and the average industrial carbon emissions and average industrial added value belong to the medium level in the eight major economic zones.

The changes in the average industrial value added and industrial carbon emissions of the eight major economic regions in China are shown in Figure 2 and Figure 3. The average industrial value added in the eastern and southern coastal regions is always the highest, and the average industrial value added in the great northwest region is always the lowest. The average industrial value added in the remaining five regions shows a slightly decreasing trend after 2016. The average industrial carbon emissions in the middle reaches of the Yellow River and the northern and eastern coast are always at the top of all regions. The average industrial carbon emissions in the Great Northwest and the Great Southwest are always less.

Furthermore, the decoupling model quantifies the relationship between industrial carbon emissions and industrial economic growth in the various provinces of China. Figure 4a shows the change in velocity decoupling distribution in various provinces in China. Due to space limitations, this paper only presents the relevant data from three long-interval years, 2000, 2009, and 2019. As can be seen from the figure, the overall velocity decoupling of Chinese provinces has proceeded from a strong decoupling to a weak decoupling to a gradual recovery of a strong decoupling. In 2019, 83.33% of China’s provinces achieved industrial carbon emission decoupling (including strong decoupling and weak decoupling), most of which were weak decoupling, while 16.67% of the provinces had not yet achieved industrial carbon emission decoupling. This shows that the industrial development of most provinces in China has entered the stage of carbon emission decoupling. Only a few provinces have achieved the best state of decoupling, while some provinces have not achieved industrial carbon decoupling and are still in the state of expansion connection or expansion-negative decoupling. Figure 4a shows the change in quantity decoupling distribution in various provinces in China. For comparison purposes, only the distribution of relevant results in 2000, 2009, and 2019 is shown here. It can be seen that the quantity decoupling state is relatively concentrated, which is mainly divided into three types: strong decoupling, weak decoupling, and expansion negative decoupling. In 2019, 86.67% of China’s provinces achieved industrial carbon emission decoupling (including strong decoupling and weak decoupling), most of which were also weak decoupling, and 13.33% of the provinces were in the state of expansion negative decoupling.

It can be seen from Figure 4, from a spatial point of view, that there are some differences in the decoupling state and decoupling process between industrial economic growth and industrial carbon emissions in China’s eight major economic zones. Nevertheless, on the whole, the velocity decoupling, quantitative decoupling state, and development process of each region are similar. The analysis combined with the decoupling index results is as follows: the velocity decoupling in the middle reaches of the Yellow River from 2007 to 2017 is in a weak decoupling state, repeatedly in other years, while quantitative decoupling is stable after 2001. Velocity decoupling and quantitative decoupling are repeatedly linked in the northwest region. Gansu and Qinghai have reached a weak decoupling state, while Ningxia and Xinjiang are still in a state of expansion negative decoupling, reflecting the fact that industrial economic growth in these areas is very dependent on energy. The northern coastal area has gradually changed from weak decoupling to strong decoupling, but there is a big difference. Beijing is in a state of strong decoupling, Hebei is in a state of expanding connection, and the northeast region has gradually changed from strong decoupling to weak decoupling. Heilongjiang Province is in a state of expansion negative decoupling.

The rest of the major regions, namely, the eastern coastal area (1998–2019), the middle reaches of the Yangtze River (2006–2019 velocity decoupling, 2002–2019 quantity decoupling), the southern coastal area (2010–2019 velocity decoupling, 2006–2019 quantity decoupling), the southwest region (2014–2019 velocity decoupling, 2012–2019 quantity decoupling) are always or after a time node in a stable weak decoupling state. This shows that although industrial carbon emissions increase with the growth of the industrial economy, the growth rate of industrial carbon emissions is lower than that of the industrial economy, and preliminary decoupling can be achieved.

The decoupling analysis method can only reflect the dependent relationship between industrial economic growth and industrial carbon emissions at the provincial level in China. However, it cannot explore the reasons for the above dependent relationship. The velocity decoupling index and quantity decoupling index for each province in China differ from the decoupling status to the development history of decoupling, so this paper further analyzes the driving factors of industrial carbon emission velocity and quantity decoupling.

### 3.2. Empirical Analysis on the Effectiveness of Decoupling Drivers

#### 3.2.1. Robustness Test

If there is a random or definite trend between variables, it may be because sometimes variables are highly correlated because they have an upward or downward trend over time at the same time and are not related, resulting in a “pseudo-regression.” In order to avoid pseudo-regression and ensure the validity of the regression estimation results, this paper uses the IPS, Hadri LM, and Fisher-PP methods simultaneously to test the stability of the original and first-order differential panel data series. Suppose that among the three methods, there are at least two methods rejecting the original hypothesis of the existence of the unit root. In that case, this series can be considered stable, and vice versa. The test results are shown in Table 3. All 11 samples are stable panel data, and the next step in regression estimation can be performed.

#### 3.2.2. Cointegration Test

Because the variables *STS*, *NP*, *GR*, and *UR* are unstable, their first-order difference is stable. Therefore, it is necessary to test further whether the series of both models are first-order stable and cointegrated. Suppose both are first-order stable and cointegration. In that case, it is proved that a long-term equilibrium relationship between the variables and subsequent regression analysis can be conducted. The test results are shown in Table 4. The long-run cointegration relationship exists, and the subsequent empirical analysis can be conducted.

#### 3.2.3. Model Selection

Considering that the random perturbation term $\mu_{it}$ is correlated with the explanatory variables, this paper has to screen between the fixed-effects regression model and the random-effects regression model. The fixed-effects regression model considers individual effects, while random effects require that all explanatory variables are uncorrelated with individual effects, i.e., the nuisance terms. In this paper, the Hausman test selects the fixed-effects model or random-effects model. The test results are shown in Table 5. Prob > chi2 for both the velocity decoupling model and the quantity decoupling model is 0.0000, both showing strong rejection of the original hypothesis, indicating that the fixed-effects model is superior to the random-effects, and the fixed-effects model regression results are accepted. Therefore, this paper chooses to establish a fixed-effects model with relevant parameters based on the results.

#### 3.2.4. Heteroscedasticity and Autocorrelation Tests

Before setting up the parameters associated with the fixed effects model, the heteroskedasticity and autocorrelation properties of the data are considered. Whether the data are homoscedastic or heteroscedastic and whether there is autocorrelation determine the difference in the parameters set for the model regression. Therefore, to set up reasonable regression models, it is necessary to test whether the data have homoscedasticity and autocorrelation. The results are shown in Table 6, and heteroscedasticity and autocorrelation exist between the two model series.

#### 3.2.5. Establish Regression Model

Based on the heteroscedasticity and autocorrelation properties of the sample, the nonparametric covariance matrix estimator is chosen in this paper. This estimation model was proposed by Driscoll et al. [52] and can obtain consistent standard errors considering heteroscedasticity and autocorrelation and applies to fixed-effects models. In Stata, the command to obtain the Driscoll & Kraay standard errors is xtscc. Therefore, this paper uses the fixed effects model with the xtscc command for the regression of the panel data. Table 7 shows the regression results.

According to the regression results in Table 3, it can be found that, first, there are differences between the regression results of the velocity decoupling model and the quantity decoupling model. 

In the velocity decoupling model, the balance of energy, the cleanliness of energy, and labor productivity are significant factors. These three indices all have negative effects on velocity decoupling. When the balance of energy, cleanliness of energy, and labor productivity increased by 1%, the velocity decoupling index decreased by 0.8%, 0.5%, and 0.006%, respectively. 

In the quantity decoupling model, the cleanliness of energy, industry structure, and population number are significant factors. The number of population and structure of the industry have positive effects on the quantity decoupling index, while the cleanliness of energy has a negative effect. This means that the quantity decoupling index increases when the number of population increases and the value added of resource-intensive industries increases. When the structure of the industry and number of population increase by 1%, the quantity decoupling index increases by 2.6% and 0.0009%.

Compared with the two models, it can be found that the cleanliness of energy has a negative effect on the two models, but the parameter is different. When cleanliness increases by 1%, the quantity decoupling index decreases by 3.6%, while the velocity decoupling index decreases by 0.5%. 

## 4. Discussion

### 4.1. Further Discussion

Unlike previous studies, this paper has the following innovations. First, in this study, the analysis and discussion are conducted from a whole industry sectoral perspective, and the differences among sectors are fully considered. Second, this study combines velocity and quantity decoupling models to comprehensively analyze the decoupling status of carbon emissions in the industrial economy. Finally, this paper constructs comprehensive and innovative driving indicators to measure the drivers of the decoupling state of carbon emissions in the industrial economy and further explores the driving mechanisms.

In 2020, the World Resource Institute [53] assessed the decoupling of carbon emissions in the United States. Results show that 41 U.S. states and Washington, D.C., representing over four-fifths of U.S. states, have decoupled their emissions from economic growth. This means that the effectiveness of the carbon decoupling mechanism is significant. A combination of technological, market, and policy initiatives have allowed states to reduce their CO_2_ emissions and increase their effectiveness in decoupling driving factors. Drivers of the mechanism are effective for various reasons, including the transition from coal to natural gas in the power sector, the rapid development of wind and solar energy, and the dominance of the economic structure in the service sector. In 2021 and 2022, the U.K. Secretary of State for Business and Energy and the French Environment and Energy Authority announced new industrial decarbonization strategies that seek to improve the effectiveness of carbon decoupling drivers. The U.K. starts with energy-intensive industries, supporting technology development to develop clean energy and reduce the carbon footprint of those energy-intensive industries. France plans to invest in carbon capture and storage technologies, from coal transition to hydrogen energy conversion, seeking to increase industrial carbon decoupling. From this paper, we can find that the effectiveness of the current carbon decoupling drivers in China is not significant. China has a long way to go to achieve complete and strong decoupling. 

### 4.2. Limitations and Future Research Direction

The research time interval of this paper is from 1998 to 2019, which is a long duration. However, due to the difficulty of obtaining indicator data and many public data that have not yet been released, this paper does not consider the data for 2020 and 2021. There is a certain degree of lag in the paper’s research comparatively. On the other hand, due to the statistical caliber, this paper takes Chinese provinces as the research object, the scale is relatively macro, and the discussion on city and sub-city levels is relatively insufficient. Based on the availability of data, future studies can include COVID-19 (2019-novel coronavirus) as a control variable or as an explanatory variable to explore the driving mechanism of industrial carbon decoupling in the context of the pandemic, which can also further explore how to maintain the stability of the decoupling status of industrial carbon emissions under unexpected circumstances. In addition, future studies can quantify the industrial carbon decoupling index and analyze its industrial carbon decoupling status from the perspective of regional perspectives, urban clusters, and some typical counties and cities to provide suggestions for sub-provincial regions to achieve industrial carbon decoupling and reach the carbon peak and carbon neutrality targets as early as possible.

## 5. Conclusions

### 5.1. Main Conclusions

This paper uses the Tapio decoupling model based on incremental data and the decoupling model based on the IPAT equation to measure the velocity and quantity decoupling relationship between industrial economic growth and industrial carbon dioxide emissions in 30 Chinese provinces for 22 consecutive years from 1998 to 2019 and uses a fixed-effects model to analyze the velocity decoupling and quantity decoupling status in 30 Chinese provinces for 17 consecutive years from 2003 to 2019. The study’s results indicate the driving factors of the decoupling.

(1) Since 1997, the average industrial added value and industrial carbon emissions have generally shown an upward trend year by year. However, there are obvious temporal and spatial differences in China’s industrial carbon decoupling. The decoupling of China’s industrial economy from the velocity and quantity of industrial carbon emissions has generally experienced a state of strong decoupling to weak decoupling. Then a slow recovery of strong decoupling. From a spatial perspective, most Chinese provinces’ velocity decoupling and quantity decoupling are in a state of weak decoupling of industrial carbon emissions. And the eight major economic zones’ industrial economic growth and industrial carbon emissions have differences in the decoupling state and process between industrial economic growth and industrial carbon emissions in the eight major economic regions.

(2) The velocity decoupling index and quantity decoupling index for each province in China have some differences from the decoupling state to the decoupling development process, and, in general, the quantity decoupling index is better than the velocity decoupling index.

(3) Reforming the structure of industrial energy and promoting labor productivity can effectively improve industrial carbon decoupling velocity. Energy consumption structure, including cleanliness and balance, significantly affects velocity decoupling. Especially, the balance of energy has a higher degree of influence on the velocity decoupling index than cleanliness. Policies for increasing the variety of energy and amount of natural gas can decrease carbon emissions when energy consumption increases. Additionally, labor productivity is also an effective influencing factor in promoting industrial carbon decoupling velocity. Companies can decrease carbon emissions by upgrading machine efficiency and training employees.

(4) Reforming industry structure and promoting energy cleanliness can effectively improve industrial carbon decoupling quantity. The cleanliness of energy and the structure of industry significantly affect quantity decoupling. The structure of the industry especially has a higher degree of influence on the quantity decoupling index than cleanliness. Besides, the number of the population also influences the quantity decoupling index through human activities. Policies supporting resource-intensive industries and promoting energy cleanliness can decrease industrial carbon emissions.

Many scholars have studied the influencing factors of carbon decoupling and demonstrated the significance of the impact of the intensity of technology investment, environmental investment, and energy consumption on carbon decoupling. For example, Wang [54] demonstrates that environmental investment and technological progress are the biggest drivers of carbon decoupling in developing countries. Zhao [55] and Jia [56] also affirm that energy intensity influences carbon decoupling in Beijing and Yunnan provinces. Liu [57] also analyzes the factors influencing carbon decoupling in different regions of China and illustrates that energy intensity significantly affects carbon decoupling. Few scholars have focused on the influence of industrial structure and labor productivity on carbon decoupling. However, the conclusions drawn in this paper differ from previous findings. Although the structural balance of energy consumption affects the velocity decoupling index, it does not affect the quantity decoupling index. Environmental investment does not affect both decoupling model indices. Instead, labor productivity and industrial structure significantly affect the speed decoupling index and the quantity decoupling index, respectively.

The conclusions of this paper differ from previous studies for three reasons. First, the shorter time dimension of the previous literature does not accurately portray the impact of changes in explanatory variables. For example, the balance of energy started to rise after China’s “dual-carbon goal” was proposed. However, provincial governments need to control the reduction of coal before reaching the carbon peak. Therefore, the energy balance changed little in recent years and cannot influentially affect the quantity decoupling index. Second, the environmental investment considered in the previous literature mainly included investments in technology and ex-post investment. The environmental investment considered in this paper is “ex-post investment,” which distinguishes governance investment from technology investment. The effect of “ex-post investment” on carbon decoupling is weak. Thirdly, in the previous literature, the industrial structure is mainly divided according to the traditional division, ignoring the possibility that there are also energy-intensive industries in light industry. In this paper, resource dependence is used as the basis for dividing industries, which can better portray the impact of industrial structure on carbon decoupling.

### 5.2. Policy Implications

Drawing on international strategies, this paper puts forward policy recommendations. For a better and faster carbon decoupling of the industrial economy, the government should:

(1) Increase procurement efforts to promote the diversification of energy products. Currently, industrial production mostly relies on fossil energy and consumes less clean energy, falling into a lack of diversification or balance of energy consumption. Now that the price level of clean energy has no obvious disadvantage over traditional fossil energy, the government can promote clean energy products through procurement. Increasing government procurement can set a good leading role for various industrial enterprises and be a way to implement subsidies. The government should increase the amount of clean energy procurement as much as possible based on its actual needs and adopt a top-down policy to stipulate the annual percentage of clean energy procurement. In addition, the government should provide relevant education and training to procurement department personnel so that they can fully realize the importance of clean energy product procurement to the economy and carbon neutrality.

(2) Establish policies to reduce the cost of clean energy and support clean energy consumption by industrial enterprises. The government can support the development of clean energy in industrial enterprises from two perspectives: fiscal policy and financial policy. From a financial point of view, the government can implement the policy of tax credits for clean energy consumption to reduce the cost of clean energy consumption. From the financial policy perspective, the government can promote financial institutions to exploit clean energy consumption loans for industrial enterprises. Financial institutions can issue consumption loans to industrial enterprises that need to purchase clean energy, support enterprises that consume high-carbon energy and promote the development of low-carbon transformation of industrial enterprises that consume high-carbon energy. 

(3) Guide enterprises to reasonably use various driving factors to promote carbon decoupling in the industrial economy. In particular, we should properly use the “double-edged sword” of the cleanliness of the energy consumption structure. The government can initiate relevant projects and establish departments to promote effective driver factors according to different enterprises and regions with different dimensions of carbon decoupling needs to achieve the ultimate goal of carbon decoupling.

In order to promote the decoupling between its industrial economic development and industrial carbon emissions, industrial enterprises should be focused on the following aspects: 

(1) Reform the energy consumption structure to meet the development needs of emission reduction. Industrial enterprises should re-examine their energy consumption structure and establish a new, low-carbon energy consumption structure system. Take energy balance, diversification, and cleanliness as guidelines for managing the energy consumption system. Furthermore, enterprises should make clean energy, mainly natural gas, wind energy, and tidal energy, as the majority of their energy consumption structure to meet their own economic development needs, while meeting the need to reduce CO_2_ emissions. 

(2) Invest in human and material resources to improve the labor-output ratio. Industrial enterprises should increase investment in human resources and introduce professional, cutting-edge talents. Enterprises should also increase business training and improve the learning curve according to the “learning by doing, learning by doing” model to improve the labor productivity of enterprises. Industrial enterprises should check and overhaul existing equipment and eliminate aging and inefficient equipment. Enterprises also should renovate and upgrade traditional equipment and other multifaceted initiatives to improve the technical level and meet the requirements for high efficiency of production.

## Figures and Tables

**Figure 1 ijerph-20-00145-f001:**
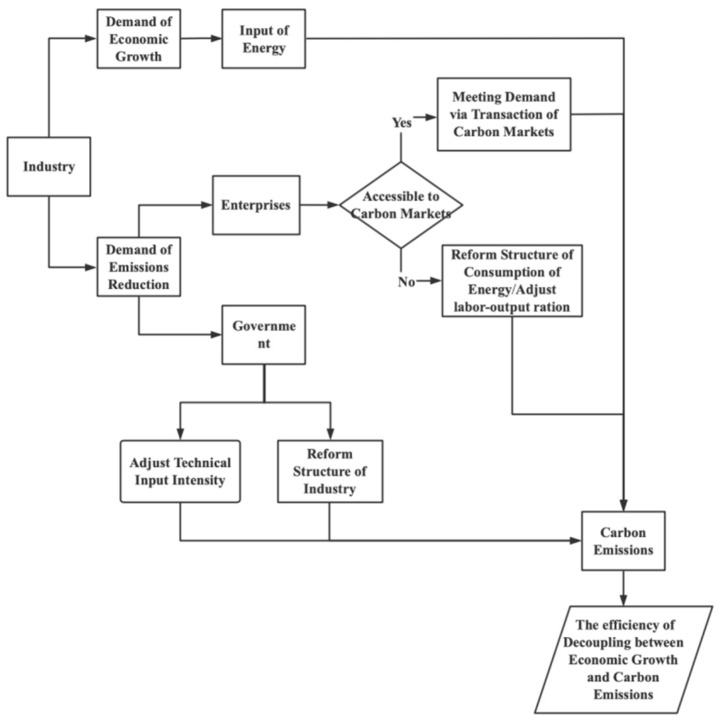
Transmission mechanism of industrial economy and carbon decoupling.

**Figure 2 ijerph-20-00145-f002:**
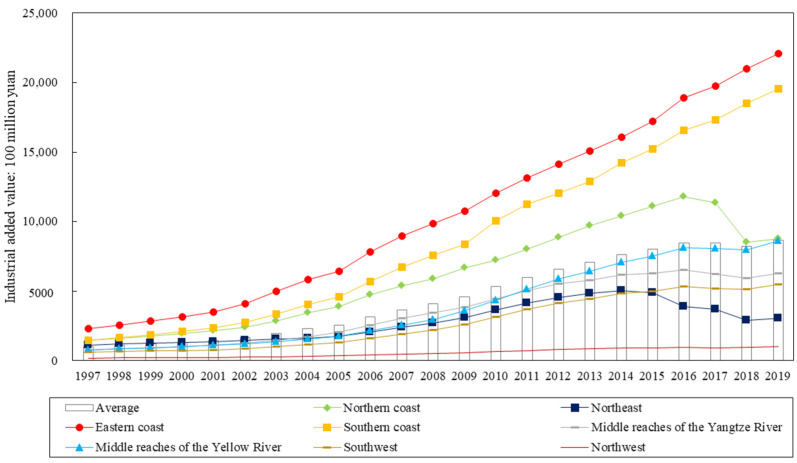
Changes in average industrial added value in China’s eight major economic zones.

**Figure 3 ijerph-20-00145-f003:**
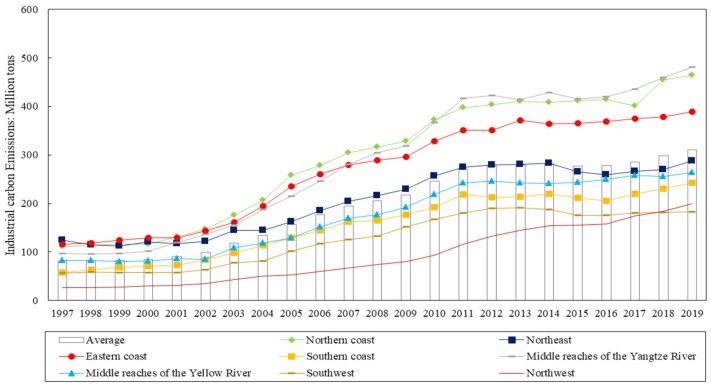
Changes in average industrial carbon emissions in China’s eight major economic zones.

**Figure 4 ijerph-20-00145-f004:**
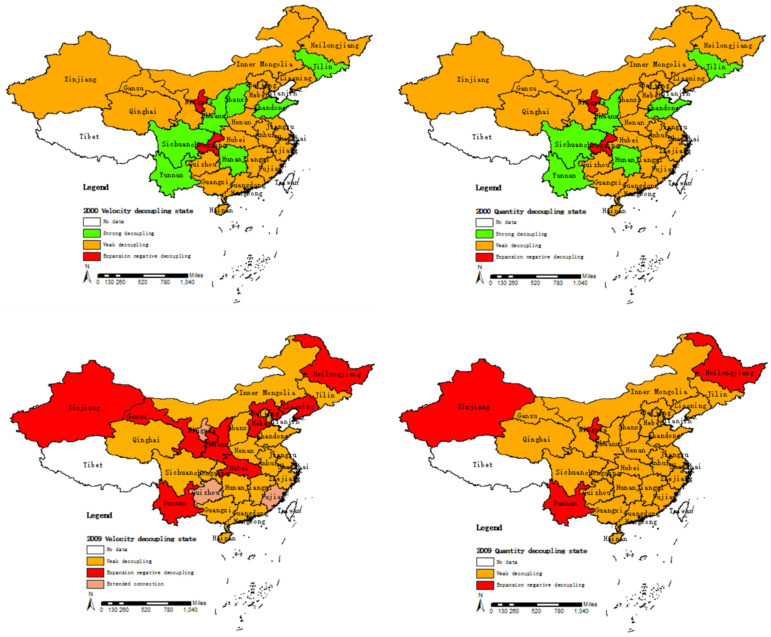
Distribution of decoupling between industrial carbon emissions and industrial economic growth in various provinces of China.

**Table 1 ijerph-20-00145-t001:** Descriptive statistics of variables.

Variable	Meaning	Numbers	Mean Value	Standard Deviation
IAD	Industrial added value	660	4701.31	1658.40
ICE	Carbon emissions of the industry	660	202.29	170.30
Ds	velocity decoupling index	510	0.59	0.48
Dq	quantity decoupling index	510	0.17	1.92
EC	Cleanliness of energy	510	0.83	0.15
EB	Balance of energy	510	0.42	0.21
LP	Productivity of labor	510	27.45	14.21
SIO	Structure of industry	510	0.37	0.19
STS	The intensity of technology investment	510	1.42	1.07
IIE	Increase investment in environmental	510	190,676.3	190,053.1
NP	Number of population	510	4470.58	2730.86
UR	Urbanization ratio	510	53.33	14.5535
GR	Government regulation	510	85.69	55.77

Note: The data are obtained from previous years of the China Statistical Yearbook, national and provincial statistical bureaus, and provincial input-output tables. Some of the data are missing and abnormal. This paper uses the trend method and regression method of conversion, and the data are substituted and processed to obtain the results of the above table.

**Table 2 ijerph-20-00145-t002:** Division of velocity decoupling and quantity decoupling state.

Division of Decoupling State		Velocity Decoupling State			Quantity Decoupling State			
		∆*I*/*I*_0_	∆G/G_0_	Ds	*g*	*t*	Relationship	*D_q_*
Negative decoupling	Expansion negative decoupling	>0	>0	(1.2, +∞)	>0	≤0	*g* > *t*	(−∞, 0)
	Strong negative decoupling	>0	<0	(−∞, 0)	<0	>0	*g* ≤ *t*	(−∞, 0)
	Weak negative decoupling	<0	<0	[0, 0.8]	<0	<0	*g* ≤ *t*	(0, 1)
Decoupling	Weak decoupling	>0	>0	[0, 0.8]	>0	>0	*g*/(*g* + 1) ≤ *t* < *g*	(0, 1)
	Strong decoupling	<0	>0	(−∞, 0)	>0	>0	*g* ≤ *t*	[1, +∞]
	Recession decoupling	<0	<0	(1.2, +∞)	<0	<0	*g*/(*g* + 1) ≤ *t* < *g*	(1, +∞)
Link	Extended connection	>0	>0	[0.8, 1.2]	>0	>0	*t* < *g*/(*t* + 1)	(0, 1)
	Decline link	<0	<0	[0.8, 1.2]	<0	<0	*t* < *g*/(*t* + 1)	(0, +∞)

**Table 3 ijerph-20-00145-t003:** Robustness test.

	Variable	Phillips-Perron Tests	Hadri LM Test	Im-Pesaran-Shin Unit-Root Test	Result
*p*-value	Ds	0.0000 ***	0.0000 ***	0.0000 ***	stable
	Dq	0.0000 ***	0.0000 ***	0.0000 ***	stable
	EB	0.0016 ***	0.0000 ***	0.0000 ***	stable
	EC	0.0001 ***	0.0000 ***	0.0000 ***	stable
	SIO	0.0002 ***	0.0000 ***	0.0000 ***	stable
	LP	0.6018	0.0000 ***	0.0003 ***	stable
	IIE	0.0030 **	0.0000 ***	0.0000 ***	stable
	STS	0.2014	0.0000 ***	0.0053 ***	unstable
	NP	0.9998	0.0000 ***	0.9871	unstable
	UR	0.3011	0.0000 ***	0.1298	unstable
	GR	0.9776	0.0000 ***	0.0254 **	unstable
	dSTS	0.0000 ***	0.0000 ***	0.0000 ***	stable
	dNP	0.0000 ***	0.0000 ***	0.0000 ***	stable
	dGR	0.0000 ***	0.0000 ***	0.0000 ***	stable
	dUR	0.0000 ***	0.0443 **	0.0000 ***	stable

Note: **, *** represent significance at the level of 5%, 1% respectively; both represented with a coefficient of 0 reject the hypothesis.

**Table 4 ijerph-20-00145-t004:** Test of cointegration relationship.

Model	Ho: No Cointegration	*p*-Value	Result
Velocity decoupling	Pedroni test for cointegration	0.0000	long-term cointegration
	Westerlund test for cointegration	0.0000	long-term cointegration
	Kao test for cointegration	0.0000	long-term cointegration
Quantity decoupling	Pedroni test for cointegration	0.0000	long-term cointegration
	Westerlund test for cointegration	0.0000	long-term cointegration
	Kao test for cointegration	0.0000	long-term cointegration

**Table 5 ijerph-20-00145-t005:** Hausman test.

Variable	Velocity Decoupling Regression Model		Quantity Decoupling Regression Model	
	Coefficient		Coefficient	
	Fixed-Effects Regression Model	Random-Effects Model	Fixed-Effects Regression Model	Random-Effects Model
EB	−0.7961285	−0.5828243	2.384521	0.3736852
EC	−0.5017271	−0.4469303	−3.578881	−1.141502
SIO	−0.6331995	−0.262268	2.592519	−1.221513
LP	−0.0060619	−0.0055131	−0.002814	0.0092246
IIE	3.35 × 10^−8^	8.92 × 10^−8^	1.26 × 10^−7^	3.18 × 10^−7^
STS	−0.0237336	−0.0642919	−0.192964	0.4952351
NP	−0.0000679	−0.0000407	0.0009458	−0.0000576
UR	−0.0039099	−0.0015072	−0.0091967	−0.0412733
GR	0.0002312	−0.0000142	0.0030897	0.0034554
	Chi2(8) = 66.44 Prob > Chi2 = 0.0000		Chi2(8) = 54.74 Prob > Chi2 = 0.0000	

**Table 6 ijerph-20-00145-t006:** Heteroscedasticity and autocorrelation test.

		Heteroscedasticity		Cross-Sectional Correlation	Serial Correlation	Result
		Breusch-Pagan Test	White’s Test	Pesaran’s Test	Wooldridge Test	
*p*-value	Velocity decoupling	0.0000	0.0000	0.0000	0.0933	Heteroscedasticity and autocorrelation exist
	Quantity decoupling	0.0007	0.0276	0.0000	0.9385	Heteroscedasticity exists; autocorrelation does not exist

**Table 7 ijerph-20-00145-t007:** Regression results of the model.

Variable	Velocity Decoupling Regression Model			Quantity Decoupling Regression Model		
	Coefficient	t	P > |t| (Robust Std) Err	Coefficient	t	P > |t| (Robust Std) Err
EB	−0.7961285	−2.04	0.050 ** (0.3894124)	2.384521	1.57	0.118 (1.521888)
EC	−0.5017271	−1.83	0.078 * (0.2747573)	−3.578881	−1.79	0.074 * (2.001074)
SIO	−0.6331995	−1.41	0.168 (0.4482697)	2.592519	2.18	0.030 ** (1.189896)
LP	−0.0060619	−1.94	0.062 * (0.0031232)	−0.002814	−0.25	0.801 (0.0111733)
IIE	3.35 × 10^−8^	0.38	0.710 (8.94 × 10^−8^)	1.26 × 10^−7^	0.17	0.864 (7.36 × 10^−7^)
STS	−0.0237336	0.37	0.711 (0.0930034)	−0.192964	−0.48	0.631 (0.4019901)
NP	−0.0000679	−0.63	0.535 (0.0001082)	0.0009458	2.38	0.018 ** (0.000398)
UR	−0.0039099	−0.57	0.571 (0.0068157)	−0.0091967	−0.40	0.687 (0.0227949)
GR	0.0002312	0.22	0.830 (0.0010686)	0.0030897	0.99	0.322 (0.0031161)
_cons	2.172607	3.18	0.003 (0.6823051)	−2.478423	−0.92	0.357 (2.68902)
	sigma_u = 0.47210629 sigma_e = 0.24561245 rho = 0.78699371			sigma_u = 2.7075269 sigma_e = 1.7438877 rho = 0.70678853		

Note: *, ** represent significance at the level of 10%, 5% respectively; both represented with a coefficient of 0 reject the hypothesis.

## Data Availability

The data used to support the findings of this study will be available from the corresponding authors upon request.

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
