# Peer review of "Driving Effect of Decoupling Provincial Industrial Economic Growth and Industrial Carbon Emissions in China"

_ijerph, 2022, doi:10.3390/ijerph20010145_

Round 1

Reviewer 1 Report

Overall it is a good article, quite interesting, but certain parts of the paper need to be restructured. I would like to make the following comments:

The position of the figure 1 should be put in a more appropriate place.

In the paragraph on line 172 a statement is made that needs to be supported by literature.

This statement in line 204 should be deleted: Professor Tapio of the Finnish Centre for Future Studies is a notable contributor to the disengagement indicator system.

A much better explanation is required as to why the regression model has been selected, explaining why a panel model was chosen and not others that can explain, such as semi-parametric or non-parametric models.

The Data Sources and Description section should be in the materials and methods sector.

The sections on Robustness Test, Cointegration Test, Model Selection, Heteroscedasticity and Autocorrelation Tests should be rewritten. The tables are in the supplementary material, but they need to appear in the body of the document so that the effects of model selection and the econometric corrections made can be understood.

Line 420, Prob>chi2 should be rewritten, referring to the tables to be incorporated in the body of the text.

On line 438, mention is made of commercial software. The research must be independent of the software tools used.

Reviewer 2 Report

In this paper, the Tapio decoupling model and IPAT-based decoupling model were used to analyze the velocity and quantity decoupling index of industrial carbon emissions in each province, and the fixed effect model was used to analyze the influencing factors of carbon decoupling. Finally, suggestions from the perspective of government and industrial enterprises were proposed. Although the presented work with valuable methodology and results deserve to be considered for publishing in the IJERPH, it still has some minor issues needed to be addressed before this step.

1. The author should cite more recent studies on this topic. for example:

Yu, X; Li, M.Q.; Kang, W.M.. Heterogeneity of Decoupling Between Economic Development and Carbon Emissions in China's Green Industrial Parks. Earths Future 2022, 10(7): 275-278.Jiang, X.H.; Raza, M.Y.. Delving into Pakistan's industrial economy and carbon mitigation: An effort toward sustainable development goals. Energy Strategy Reviews 2022, 41 :100839.

2. In section 3.1.1, describe clearly the source of industrial carbon emission, database, and calculate the equation.

3. If possible, please add a graphical abstract.

4. Section 5.1's first conclusion is similar to the results and analysis in 3.1.2.

5. Some formatted errors, like spacing, must be checked thoroughly.

Reviewer 3 Report

This article is interesting and closed related with the current situation in China. However there are still need some improvements.

1. Sentence in line 27-28 seemed awkward. Please revise it more appropriately.  

2. The way on writing citations are not consistent.

3. From the previous studies that authors wrote, which one use the panel data?

4. Why do the authors not include the increase of investment in the hypotheses?

5. Why does the study not consider government regulation as explanatory variable?

6. The method is very long. It is better to re-arrange the writing.

7. The article does not explain about panel data analysis in the method

8. It would be more clearly if the author presents the mapping  of industrial type that related to Figure 3. What kind of industry in in eastern coast, northern coast, etc

9. I cannot see the note of legend on the figure 4 clearly.

10. Why this study used regression analysis not panel data analysis?

11. Some important indicator statistics are not presented in the Table 3

12. Compare the result with previous studies on the discussion

Reviewer 4 Report

Driving Effects of Decoupling between Industrial Economic Growth and Industrial Carbon Emissions —A Research Based on Panel Data of 30 Provinces in China

by Jingfen Hua , Junli Gao, Ke Chen and Jiaqi Li

Review

To estimate the industrial carbon emissions of 30 provinces in China, the authors use the data between 1998 and 2019 and analyze each province’s velocity and quantity decoupling index of industrial carbon emissions.

The results show that the industrial carbon emissions increase yearly and there are significant differences among provinces. The carbon decoupling of the industrial economy in most provinces is weak, and the quantitative decoupling index is better than the velocity decoupling index.

The manuscript is interesting and the topic is important.

The presentation of state of the art identifies significant publications.

The letters in Figures 4 a and b are not clear. The paragraphs are very large, could be fragmented to help the reader.

Specific recommendations should be added in section Conclusions.

The title is long; a more compact formulation is recommended.

Reviewer 5 Report

This research paper explores an important dimension of environmental quality which is needed but there are some additional observations to improve the quality of the paper.

In the introduction, authors need to add some theoretical relationship among decoupling and industry. Moreover, some numerical evidences in this regard will signify the study. These numerical relationship may be draw graphically for better understanding of their trends and ongoing relationship.

It is also important to highlight the novelty of the study.

By incorporating a separate section of literature review on the topic will facilitate the readers about debating trend for decoupling and industry.

There is need to add the rationale for selection of sample and time period.

 Authors are suggested to highlight some merits of Tapio decoupling model.

The discussion section should be amended by linking the findings of earlier studies on the topic. If findings are different form earlier literature then strong arguments should be there. 

Round 2

Reviewer 1 Report

All of my corrections made by author(s). It is appropriate for me to publish the article as it is.

Reviewer 5 Report

The authors have incorporated the comments.